# Stimbiotics Supplementation Promotes Growth Performance by Improving Plasma Immunoglobulin and IGF-1 Levels and Regulating Gut Microbiota Composition in Weaned Piglets

**DOI:** 10.3390/biology12030441

**Published:** 2023-03-13

**Authors:** Wenning Chen, Chenggang Yin, Jing Li, Wenjuan Sun, Yanpin Li, Chengwei Wang, Yu Pi, Gustavo Cordero, Xilong Li, Xianren Jiang

**Affiliations:** 1Key Laboratory of Feed Biotechnology of the Ministry of Agriculture and Rural Affairs, Institute of Feed Research, Chinese Academy of Agricultural Sciences, Beijing 100081, China; 2College of Life Science, Jiangxi Science and Technology Normal University, Nanchang 330013, China; 3AB Vista, Marlborough SN8 4AN, UK

**Keywords:** growth performance, immunity, microbiome, piglets, stimbiotics

## Abstract

**Simple Summary:**

Recently, stimbiotics (STB) have been suggested as a non-digestible and fermentable additive that can improve microbial fibre fermentation. This study observed that STB could improve immunity and IGF-1 levels in plasma, regulate the faecal microbial community, and have a positive effect on weaned piglets’ feed intake and daily weight gain. Based on the results of this study, weaned piglets’ growth performance was highly improved with STB inclusion in the diet, and this could provide a basis for its use in pig production.

**Abstract:**

This study was conducted to investigate the effects of dietary supplementation with stimbiotics (STB) on growth performance, diarrhoea incidence, plasma antioxidant capacity, immunoglobulin concentration and hormone levels, and faecal microorganisms in weaned piglets. Compared with the control (CT) group, the addition of STB improved the body weight (BW) of piglets on days 28 and 42 (*p* < 0.05) and increased daily weight gain and daily feed intake from days 14–28 and throughout the trial period (*p* < 0.05). Correspondingly, the plasma insulin-like growth factor 1 (IGF-1) level on day 42 was significantly improved by STB (*p* < 0.05). VistaPros (VP) group levels of immunoglobulin (Ig) A and G were significantly higher on days 14 and 42 (*p* < 0.05) than the CT group levels. In addition, the activity of plasma catalase tended to be increased on day 14 (*p* = 0.053) in the VP group, as for superoxide dismutase, glutathione peroxidase, and malondialdehyde, STB did not significantly affect their levels (*p* > 0.05). Moreover, dietary STB increased the relative abundance of beneficial bacteria, including *norank_f_Muribaculaceae*, *Rikenellaceae_RC9_gut_group*, *Parabacteroides*, and *unclassified_f__Oscillospiraceae*. In summary, STB improved the immunity and IGF-1 levels in the plasma of weaned piglets and consequently promoted the growth performance of weaned piglets.

## 1. Introduction

During the early weaning period, the immaturity of piglets’ organs, coupled with changes in diet, nutritional sources, environment, and psychology, can cause strong stress reactions in piglets. Among them, piglets’ dietary habits transfer from sow milk to corn-soybean meal-based diets. The presence of some anti-nutritional factors in the diet can have an impact on the growth performance of weaned piglets.

Non-starch polysaccharides (NSP) are anti-nutritional factors. Maize-soybean meal basal diets include large amounts of NSP present in the plant cell wall, mainly arabinoxylan (AX) in maize [1], and xyloglucan and xylan in soybean meal [2]. Many previous studies reported that NSP could enclose intracellular nutrients, increase the viscosity of the digestive diet, alter the morphology of the intestinal epithelium, and reduce nutrient digestibility [3,4,5,6], then influence growth performance and decrease carcass yield [7,8]. However, NSPs are also an important nutrient for monogastric animals, especially for gut bacteria fermentation, which depends on the NSP components. It has been reported that exogenous addition of non-starch polysaccharide degrading enzymes (especially xylanase, XYL) can degrade insoluble and large soluble polysaccharides in the diet to produce oligosaccharides [9,10], oligosaccharides can be well fermented by intestinal bacteria. In addition, xylo-oligosaccarides (XOS) with a degree of polymerization in the range of 2–6, were identified to improve intestinal health and stimulate immune responses in animals, acting in a prebiotic manner [11].

Recently, a new class of feed additives has been proposed, namely, stimbiotics (STB), which are a complex of XYL and XOS [12]. STB is capable of hydrolyzing arabinoxylans and also providing short oligosaccharides. Thus, STB may serve the dual purpose of reducing the anti-nutritional effects of NSP while also stimulating SCFA production by the microbiota. Craig et al. (2020) reported that the addition of STB reduced digestive viscosity and increased the concentration of SCFAs [13]. This is very relevant to reduce the negative effects of weaning stress in piglets and promoting growth performance.

We hypothesized that supplementing STB with a maize-soybean meal-based diet would reduce the anti-nutritional effects of NSP and provide additional prebiotics for beneficial bacteria, release more energy from the feed, improve intestinal health and immunity, which would result in the improvement of the growth performance, remission of weaning stress, and reduction of the diarrhoea incidence of weaned piglets. Thus, the purpose of this study was to examine the effect of STB addition to the diet affects growth performance, diarrhoea incidence, antioxidant capacity, immunity, and intestinal microbiota in weaned piglets.

## 2. Materials and Methods

### 2.1. Animals and Experimental Design

The trial took place at the Tianpeng experimental farm, located in Langfang. We randomly distributed 80 healthy weaned piglets (Duroc × Landrace × Yorkshire) with little difference in body weight (BW, 8.84 ± 0.26 kg) and age (28 ± 1 day) to two treatment groups. A basal diet was fed to the control (CT) group, and a basal diet supplemented with 100 g/t of VistaPros (STB, mainly containing β-1,4-endo-xylanase and xylo-oligosaccharides, was supplied by AB Vista, Marlborough, Wiltshire, UK.) was fed to the VistaPros (VP) group. Each group had 8 replicates (pens), each replicate had 5 pigs, and the trial period was 42 d, divided into 2 stages (pre-nursery and post-nursery), with 2 kg/t of ZnO added to the diets of both treatment groups in the pre-nursery period. The piglet basal diet was formulated according to NRC (2012) requirements for piglets in the nursery stage (Table 1). The experiment was conducted in a fully enclosed nursery house, and all the piglets were kept in pens (5 pigs/pen), and fed and watered ad libitum. The starting temperature in the pigsty was 28 degrees Celsius, then decreased by 1 °C every week until the final temperature reached 25 degrees Celsius. The pens were inspected daily, and the pens were cleaned periodically.

### 2.2. Sample Collection

On days 14 and 42 of the experiment, one piglet was arbitrarily taken from each pen, and a blood sample was taken from the pig’s jugular vein into a heparin tube and centrifuged at 3000 r/min for 20 min. The plasma was stored at −20 °C for analysis of antioxidant and immunological parameters and insulin-like growth factor 1 (IGF-1) levels. On day 42, faecal samples (approximately 20 g) were collected from one randomly selected pig from each pen by rectal massage. The samples were stored frozen at −80 °C for later analysis. 

### 2.3. Growth Performance and Diarrhoea Incidence Measurements

On day 0 of the trial, the piglets were weighed individually and grouped by weight. However, on days 14, 28, and 42 of the trial, piglets were weighed according to each pen. The remaining feed was weighed at the time of weighing and the average feed consumption was calculated for each period based on our recorded diet consumption from day 0 to days 14, 28, and 42 The growth performance of weaned piglets was evaluated by calculating average daily gain (ADG), average daily feed intake (ADFI), and feed conversion ratio (FCR) for each replicate. All culled or dead piglets were also recorded daily and feed consumption was corrected accordingly. The incidence of diarrhoea was assessed by a five-point stool concentration scoring system: 1 = hard, dry pellet; 2 = firm, formed stool; 3 = soft, moist feces that retains its shape; 4 = soft, unformed feces; and 5 = watery liquid that can be poured. A liquid consistency (score 4–5) was considered indicative of diarrhoea [14,15]. The incidence of diarrhoea (%) was calculated as a percentage of the number of piglets with diarrhea divided by the total number of piglets in each treatment.

### 2.4. Assay of Plasma Antioxidant Indicators

The plasma antioxidant indices include glutathione peroxidase (GSH-Px), superoxide dismutase (SOD), catalase (CAT), and malondialdehyde (MDA). Using 5,50-dithiobis-*p*-nitrobenzoic acid to analyze GSH-PX with an absorption peak at 412 nm. SOD activity was measured by the WST-1 method, which uses a water-soluble thiazole salt (WST-1) to react with a superoxide anion to produce a water-soluble dye, thereby measuring SOD activity with an absorption peak at 450 nm. We measured CAT activity with ammonium molybdate and its absorption peak was then 405 nm. Measurement of MDA activity by the TBA method, which condenses MDA with 2-thiobarbituric acid to form a red product with an absorption peak at 532 nm. The above kits were purchased from Nanjing Jiancheng Bioengineering Institute (Nanjing, China), and the determination was carried out according to the kit instructions.

### 2.5. Assay of Plasma Immune Markers and IGF-1

Immune indicators include Immunoglobulin A (IgA) and Immunoglobulin G (IgG). According to the manufacturer’s instructions, ELISA kits are used to determine plasma IgA and IgG (Jiancheng Bioengineering Institute, Nanjing, China). The kit determines IgA and IgG in plasma using a competitive method with an absorption peak at 450 nm. According to different dilution concentrations of recombinant porcine immunoglobulin, ELISAcalc was used to fit the logistic curve (four-parameter equation) and calculate the content. The results were expressed in mg/mL. IGF-1 levels were determined using an ELISA kit (mlbio, Shanghai, China), and the determination was carried out according to the kit instructions. The ELISA kit uses a double antibody sandwich method to determine the level of IGF-1 in the sample and measures the absorbance at 450 nm to calculate the level of IGF-1 from the standard curve. The intra-assay and the inter-assay coefficients of variation for ELISA in the samples were less than 10% and 15%, respectively.

### 2.6. Faecal Microbial Composition Analysis

The faecal samples were frozen at −80 °C and sent to Major bio (Shanghai, China) for DNA extraction and 16 S rRNA gene sequencing was performed by high-throughput sequencing technology. DNA concentration was measured using a Nanodrop-1000 instrument (Thermo Scientific, Waltham, MA, USA) and DNA quality was assessed by 1% agarose gel electrophoresis. Based on the hypervariable region of the bacterial 16 S rRNA gene V3–V4, barcode fusion primers (338F: 5-ACTCCTACGGGAGGCAGCAG-3, 806R: 5-GGACTACHVGGGTWTCTAAT-3) were used to amplify the extracted genomic DNA. PCR products were detected on 2% agarose gel electrophoresis and recovered by cutting the gels using the AxyPrepDNA Gel Recovery Kit (Axygen Biosciences, Union City, CA, USA). Referring to the preliminary quantification results of electrophoresis, the PCR products were quantified by using QuantiFluor™-ST Blue Fluorescence Quantification System (Promega, Madison, WI, USA). The pure PCR products were used to build Miseq libraries and sequenced using the MiSeq platform (Illumina, San Diego, CA, USA) on Major Bio (Shanghai, China). Categorize unique sequences into identical amplicon sequence variants (ASVs) by a 99% similarity threshold. Statistical analysis was performed using the ASV agglomerative results.

### 2.7. Statistical Analysis

Data were analysed as a randomized complete block design using the GLM procedure of SAS 9.2, and multiple comparisons were performed by the Tukey HSD method. In this model, the experimental unit for data related to growth performance is the pen, and the experimental unit for data related to plasma indicators is the individual piglet. Furthermore, piglet diarrhoea rates were analysed using a chi-square test. Bioinformatic analysis of the faecal microbiota was carried out using the Majorbio Cloud platform (https://cloud.majorbio.com, accessed on 8 February 2023). Alpha- and beta-diversity were determined with Mothur (v 1.30.1) and Qiime2 (v2022.2), respectively, and counted and plotted with R software (v 3.3.1). Principal coordinate analysis (PCoA) was used to assess beta-diversity and ANOSIM statistical assay was used to determine significant differences in beta-diversity. Venn diagrams enable the calculation of the number of common and unique species (e.g., ASVs) in multiple populations or samples and provide a more visual representation of the similarity and overlap of species (e.g., ASVs) composition in environmental samples. The rank-sum test used in the differential species analysis was a nonlinear model (GLM model: generalized linear model). It was considered significant when *p* < 0.05 and propensity when 0.05 < *p* < 0.10.

## 3. Results

### 3.1. Growth Performance and Diarrhoea Incidence

Table 2 shows the effect of STB addition on growth performance and incidence of diarrhea in weaned piglets. Piglets in the VP group had significantly higher body weights on days 28 and 42 (*p* < 0.05) and significantly higher ADG and ADFI during days 14–28 and 0–42 (*p* < 0.05) compared to those in the CT group. Yet neither FCR nor diarrhoea incidence differed significantly between CT and VP (*p* > 0.05).

### 3.2. Plasma Antioxidant Capacity

The effect of STB in the diet on the antioxidant capacity of weaned piglets is presented in Table 3. There was no significant effect of STB on the antioxidant capacity of weaned piglets (*p* > 0.05). On day 14, there was a tendency for plasma CAT activity to be higher in the VP group than in the CT group (*p* = 0.053), while dietary STB supplementation did not significantly affect SOD, GSH-PX, and MDA levels (*p* > 0.05). Further, the CT and VP groups did not show significant differences on day 42 when it came to plasma CAT, SOD, GSH-PX, and MDA (*p* > 0.05).

### 3.3. Plasma Immunoglobulin and IGF-1 Levels

Table 4 shows the effects of STB on plasma immunoglobulins of weaned piglets. Plasma IgA and IgG levels were significantly higher in the VP group compared to those in the CT group on days 14 and 42 (*p* < 0.05). The effect of STB in the diet on IGF-1 levels in the plasma of weaned piglets is shown in Figure 1. Plasma IGF-1 levels were significantly higher in the VP group than in the CT group at day 42 (*p* < 0.05), and they were not significantly different from the CT group at day 14 (*p* > 0.05).

### 3.4. Analysis of Faecal Microorganisms

The Venn diagram was performed based on the ASVs counted and showed that there was a total of 2347 ASVs obtained, of which 1040 ASVs (about 44%) co-located between the CT and VP groups, and 671 and 636 ASVs uniquely presented in the CT and VP groups respectively (Figure 2A). The differences in faecal microbial community diversity between the CT and VP groups were analysed and the α and β indices were estimated. Regarding alpha diversity (Table 5), the richness and diversity of species in the microbial community were obtained by assessing a series of Alpha diversity indices. Sobs, Chao, and Ace indices indicated richness, and Shannon and Simpson’s indices indicated diversity. No significant differences were found in the Sobs, Chao, and Ace indices (*p* > 0.05) in the VP group compared with the CT group, nor in the Shannon and Simpson indices (*p* > 0.05). Regarding β-diversity (Figure 2B), the more similar the community composition of the samples was based on the Bray-Curtis distance PCoA analysis, the closer they were in the PCoA map. As shown in Figure 2B, the CT and VP groups were closer, indicating that the microbial community composition was similar between the two treatment groups.

At the phylum level (Figure 3A), *Firmicutes*, *Bacteroidota*, and *Spirochaetota* were the major dominant phylum. The relative abundance of Firmicutes was 84.00% and 81.33% in the CT and VP groups, respectively, and the relative abundance of *Bacteroidota* was 10.97% and 13.39% in the VP group, respectively. The relative abundance of *Spirochaetota* in CT and VP was 3.36% and 3.66%, respectively. At the genus level (Figure 3B), the top five microorganisms in relative abundance in the CT group were *Clostridium_sensu_stricto_1* (24.56%), *Terrisporobacter* (6.37%), *Christensenellaceae_R-7_group* (5.19%), and *Treponemanorank* (4.48%) and *norank_o__Clostridia_UCG-014* (4.35%). VP group was mainly dominated by *Clostridium_sensu_stricto_1* (33.23%), *Terrisporobacter* (6.10%), *Christensenellaceae_R-7_group* (5.62%), *unclassified_f__Lachnospiraceae* (4.08%) and *UCG-005* (3.46%).

The Wilcoxon rank-sum test was used to analyse the microbial communities of the samples for significant differences at the phylum level and genus level. At the phylum level (Figure 4A), the relative abundances of Actinobacteriota and Desulfobacterota were significantly decreased in the VP group compared to those in the CT group (*p* < 0.05). At the genus level (Figure 4B), compared with the CT group, the relative abundance of *norank_f__UCG-010*, *norank_f__norank_o__Bradymonadales* and *norank_f__Paludibacteraceae* significantly increased in relative abundance, and *Olsenella*, *norank_f__norank_o__Bacteroidales*, *Bacteroides_pectinophilus_group*, *Senegalimassilia*, and *Mucispirillum* significantly decreased in relative abundance (*p* < 0.05), *g__norank_f__Muribaculaceae*, *g__Rikenellaceae_RC9_gut_group*, *g__Parabacteroides*, *g__unclassified_f__Oscillospiraceae* tended to increase in relative abundance (*p* = 0.08), and *g__Lachnospiraceae_FCS020_group* tended to decrease in relative abundance (*p* = 0.09).

## 4. Discussion

Recent studies reported that STB could be used as a supplement in the diets of monogastric animals to improve their growth performance, reduce the inflammatory response and enhance gut health, especially in young animals [16,17], as a result of the functions of STB to accelerate fibre fermentation and produce SCFAs in the hindgut [18]. In the present study, we demonstrated that the supplementation of STB in the diet significantly improved body weight, daily weight gain, and daily feed intake of weaned piglets, which was in line with earlier findings that STB significantly improved the growth performance of monogastric animals [17,18]. It is noteworthy that there was no significant difference in FCR, which may be due to the short period of our experiment, and the changes in FCR were not reflected. However, the IGF-1 levels in piglets were improved, which was consistent with the changes in body weight, daily weight gain, and daily feed intake of weaned piglets. IGF-1 is a central hormone in growth regulation and is closely related to nutritional status [19], and malnutrition significantly reduces serum IGF-1 levels and can return to normal concentrations after re-feeding [20]. In the present study, the IGF-1 levels of weaned piglets were significantly elevated by STB, possibly due to increased dietary energy and protein intake, as reduced IGF-1 occurs when diet or protein intake is insufficient [21,22]. Weaning usually induces impaired growth and reduced serum IGF-1 levels in piglets [21,22], which are significantly associated with reduced feed intake [23]. The current study showed that adding STB to the diet can significantly improve the feed intake of weaned piglets in the late nursery period, and serum levels of IGF-1 on day 42 were also significantly increased, suggesting a positive correlation between increased IGF-1 levels and growth performance, which is consistent with the literature [24].

The mucosal immune system is not fully mature when piglets are weaned, are also encounter psychosocial, physical, and nutritional stressors that can have a negative cumulative effect on the immune response, resulting in lagging growth after weaning. It has been demonstrated that the plasma IgA and IgG concentrations will decrease directly after weaning [25,26], causing severe weaning stress [27]. The main functions of IgA and IgG are respectively to protect mucosal surfaces from infectious microorganisms [28] and recognize antigens on the surface of invading viruses and bacteria and recruit effector molecules [29], playing a key role as antibodies against infection in the immune system [30,31,32]. Our study demonstrated that weaned piglets supplemented with STB had substantially higher levels of immunoglobulins (IgA and IgG) than the CT group, suggesting that dietary STB can improve the immune system of piglets. However, STB had no obvious effect on reducing the rate of diarrhoea in weaned piglets, and the main reason might be related to the addition of ZnO to the pre-stater diet. It is widely known that feeding ZnO for 14 consecutive days after weaning is effective in preventing and reducing the incidence of diarrhoea in weaned piglets [33]. Therefore, low diarrhoea incidence was obtained in the two groups during the early nursery period.

The composition of the bacterial community in the gut plays an important role in intestinal health and systemic growth [34], and the composition of the microbial community is strongly influenced by the composition of the diet and environmental factors. The intestine in piglets contains a wide variety of microorganisms, more than 90% of which are *Firmicutes* and *Bacteroidetes*, which play an important role in improving the immunity of the body and nutrient absorption and metabolism [35]. Therefore, having a stable intestinal microflora structure is one of the important factors for the healthy growth and development of animals. In this study, *Firmicutes*, *Bacteroidota*, and *Spirochaetota* were found to be the main dominant phylum in piglets’ faeces, and the sum of the relative abundances of both *Firmicutes* and *Bacteroidota* was greater than 90%, while the absence of obvious differences between the CT and VP groups, showing that STB could maintain the balance of faecal microorganisms at the phylum level. In addition, at the genus level, it was found that *Clostridium_sensu_stricto_1* and *unclassified_f_Lachnospiraceae* were the dominant bacteria in the VP group. The VP group also increased the relative abundance of *norank_f_Muribaculaceae*, *Rikenellaceae_RC9_gut_group*, *Parabacteroides*, and *unclassified_f__Oscillospiraceae*, of which *Muribaculaceae* were positively associated with the barrier function of the intestinal mucus layer and play a role in the degradation of complex carbohydrates that can produce acetic acid and propionic acid [36,37]. *Rikenellaceae_RC9_gut_group* was involved in the degradation of polysaccharides of plant origin [38], indicating more fermentable fiber content in the large intestine used by bacteria [39]. Moreover, *unclassified_ f_Lachnospiraceae* had a positive correlation with ADFI in weaned piglets, which was consistent with the increased ADFI in growth performance [40]. *Clostridium_sensu_stricto_1*, *Parabacteroides*, and *unclassified_f__Oscillospiraceae* were also the main producers of SCFAs by the beneficial bacteria [41,42,43], all of which demonstrated that the supplement of STB could increase fibre fermentation and SCFAs production in the large intestine of the piglet. Thus, STB has a potentially beneficial effect by promoting the growth of beneficial bacteria to utilize dietary fibre and possibly increase SCFAs production that helps weaned piglets to obtain energy from the ration and thus improve their growth performance and immune system. Unfortunately, since fecal samples are more readily available, we only collected feces for microbial analysis, and the fecal microbial community may be very different from that in the colon or even the gut, also the relative abundance of these differential microorganisms is low, so it cannot be verified that all of these microorganisms are at play, so more specific studies will be conducted later.

Weaning is a rather challenging and stressful event for pigs, which can disrupt the physiological balance of oxidants and antioxidants and lead to oxidative stress [44,45]. Previous studies reported the effects of two major parts of STB on the antioxidant capacities in weaned piglets, xylanase could increase the total antioxidant capacity and decrease the concentration of MDA in weaned piglets [46,47], and XOS significantly increased SOD and CAT capacities and decreased MDA level in weaned piglets [48]. However, in the present study, there was no significant difference in the levels of plasma MDA, SOD, and GSH-Px on day 14 and day 42 in the weaned piglets with STB addition though there was a trend of increased plasma CAT activity in weaned piglets supplemented with STB on day 14, which might be related to the limited stress on the piglets. All weaned piglets used in this trial were directly transferred from the same farm without long-distance transportation, and piglets in the farrowing house and nursery house had similar environmental microorganisms and feeding management, which resulted in reduced external stimuli to the piglets. Song et al. (2022) demonstrated that STB addition to the diet could alleviate the negative effects on growth performance and gut health when administered Shiga toxigenic Escherichia coli (STEC) orally to weaned piglets [17]. Thus, further experiments should be conducted under challenging conditions to confirm the effect of STB on the antioxidant capacity of weaned piglets.

## 5. Conclusions

The growth performance of weaned piglets improved significantly after STB was added to the diet, probably due to improved immunity and IGF-1 levels in piglets. As STB supplementation provides piglets not only with enzymes that degrade non-starch polysaccharides but also with prebiotics that benefit intestinal health and immunity, which are used by microorganisms in the intestine to exert helpful effects on the piglets. This is significant for piglets to utilize the nutrients in the feed and improve growth performance.

## Figures and Tables

**Figure 1 biology-12-00441-f001:**
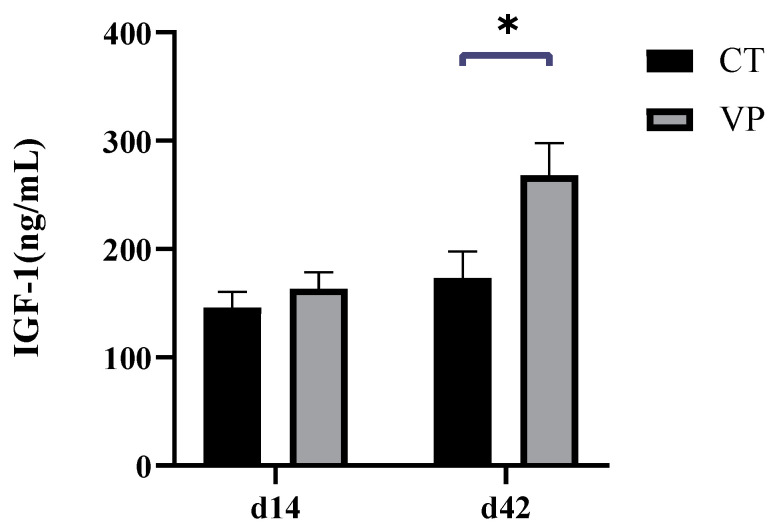
Effect of dietary supplementation with stimbiotics on plasma hormone levels of weaned piglets, the sign *—indicates the degree of significant difference, * *p* < 0.05.

**Figure 2 biology-12-00441-f002:**
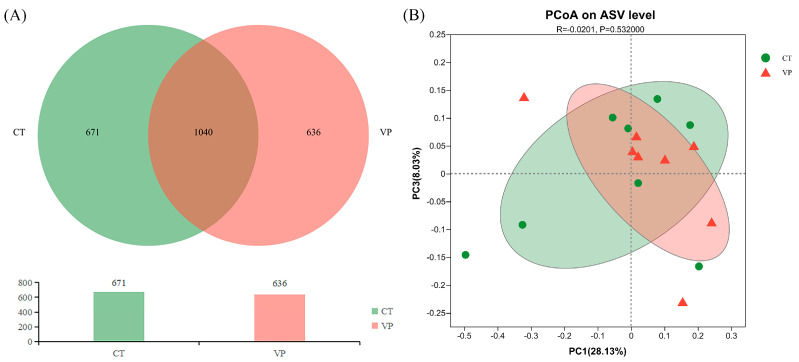
Effects of dietary supplementation with stimbiotics on faecal microbial communities. (**A**) ASVs Venn diagram. (**B**) Faecal microbial β−diversity based on the ASVs level.

**Figure 3 biology-12-00441-f003:**
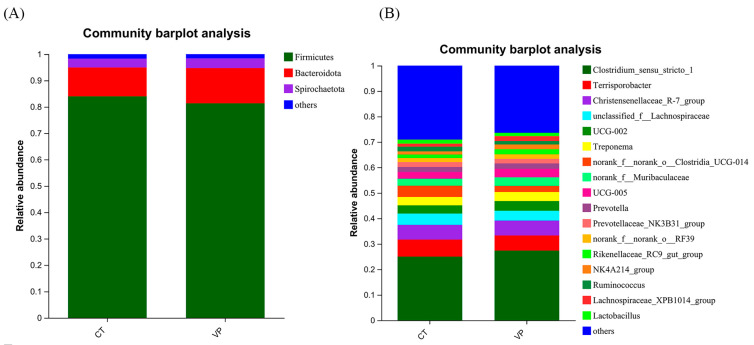
(**A**) The relative abundance of faecal microbiota composition at the phylum level, the *Y*-axis represents the average relative abundance, the *X*-axis represents the different groups, and the columns with different colors represent different groups. (**B**) The relative abundance of fecal microbial genus, the *Y*-axis represents the average relative abundance, the *X*-axis represents the different groups, and the columns with different colors represent different groups.

**Figure 4 biology-12-00441-f004:**
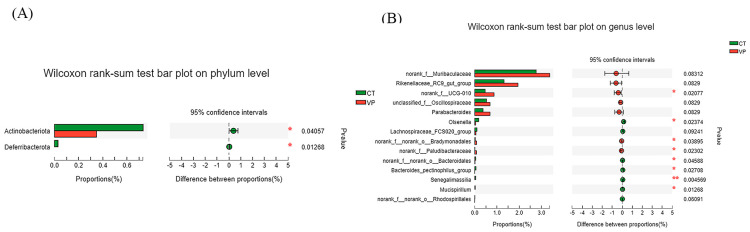
(**A**) Significance test of the difference between treatments at phylum. (**B**) Significance test of the difference between treatments at genus levels, the *Y*−axis represents the species names at a certain taxonomic level, the *X*−axis represents the average relative abundance in different groups of species, and the columns with different colors represent different groups; the far right is the *p*−value, * *p* < 0.05, ** *p* < 0.01.

**Table 1 biology-12-00441-t001:** Ingredient and calculated nutrient composition of the basal diet (as fed basis, %).

Items	Pre-Starter (0–14 Days)		Starter (14–42 Days)	
	CT	VP	CT	VP
Ingredients				
Corn	16.45	16.45	21.17	21.17
Extruded corn	32.00	32.00	40.00	40.00
Soybean meal	14.00	14.00	17.50	17.50
Extruded soybean	11.50	11.50	6.00	6.00
Fish meal	5.60	5.60	3.00	3.00
Whey	15.00	15.00	5.00	5.00
Soybean oil	1.00	1.00	1.20	1.20
Bran	-	-	1.50	1.50
Dicalcium phosphate	0.40	0.40	0.60	0.60
Limestone (CaCO_3_)	0.75	0.75	0.90	0.90
Salt	0.30	0.30	0.30	0.30
Choline chloride (60%)	0.05	0.05	0.05	0.05
L-Lysine HCl	1.20	1.20	1.08	1.08
DL-Methionine	0.09	0.09	0.08	0.08
Threonine	0.27	0.27	0.24	0.24
Tryptophan	0.02	0.02	0.01	0.01
Phytase	0.02	0.02	0.02	0.02
Acidifier	0.35	0.35	0.35	0.35
Vitamin and mineral premix ^1^	1.00	1.00	1.00	1.00
Total	100.00	100.00	100.00	100.00
Analysed nutrient content, %				
Crude protein	18.63	18.34	16.90	17.06
Calcium	1.02	0.91	1.13	0.98
Phosphorus	0.62	0.64	0.55	0.58
Ether extract	5.63	5.82	6.14	5.77
Calculated nutrient content, %				
ME, MJ/kg	14.23	14.23	14.02	14.02
Lysine	1.30	1.30	1.15	1.15
Methionine	0.38	0.38	0.34	0.34
Threonine	0.76	0.76	0.68	0.68
Tryptophan	0.21	0.21	0.19	0.19

^1^ Premix supplied per kg of diet: niacin, 38.4 mg; calcium pantothenate, 25 mg; folic acid, 1.68 mg; biotin, 0.16 mg; vitamin A, 35.2 mg; vitamin B1, 4 mg; vitamin B2, 12 mg; vitamin B6, 8.32 mg; vitamin B12, 4.8 mg; vitamin D3, 7.68 mg; vitamin E, 128 mg; vitamin K3, 8.16 mg; zinc (ZnSO_4_·H_2_O), 110 mg; copper (CuSO_4_·5H_2_O), 125 mg; selenium (Na_2_SeO_3_), 0.19 mg; iron (FeSO_4_·H_2_O), 171 mg; cobalt (CoCl_2_), 0.19 mg; manganese (MnSO_4_·H_2_O), 42.31 mg; iodine (Ca(IO_3_)_2_), 0.54 mg.

**Table 2 biology-12-00441-t002:** Effect of dietary supplementation with stimbiotics on growth performance and diarrhoea incidence of weaned piglets.

Items	Treatment		SEM	*p*-Value
	CT	VP		
BW, kg				
Day 0	8.84	8.84	0.38	1.000
Day 14	10.85	11.22	0.39	0.178
Day 28	15.41	16.70	0.57	0.013
Day 42	20.52	22.25	0.78	0.031
ADG, g				
Days 0–14	144	171	14	0.171
Days 14–28	325	391	22	0.022
Days 28–42	365	396	24	0.399
Days 0–42	278	319	14	0.041
ADFI, g				
Days 0–14	266	291	15	0.331
Days 14–28	629	711	29	0.025
Days 28–42	728	807	41	0.200
Days 0–42	541	603	20	0.038
FCR				
Days 0–14	1.89	1.80	0.12	0.450
Days 14–28	1.98	1.83	0.07	0.193
Days 28–42	2.03	2.05	0.10	0.893
Days 0–42	1.97	1.89	0.05	0.384
Diarrhoea incidence, %				
Days 0–14	1.96	2.55	-	0.514

**Table 3 biology-12-00441-t003:** Effect of dietary supplementation with stimbiotics on plasma antioxidant capacity of weaned piglets.

Items	Treatment		SEM	*p*-Value
	CT	VP		
Day 14				
CAT/(U/mL)	1.59	2.36	0.26	0.053
SOD/(U/mL)	23.81	25.09	1.49	0.575
GSH-PX/(U/mL)	253	263	14	0.648
MDA/(nmol/mL)	3.49	3.40	0.15	0.730
Day 42				
CAT/(U/mL)	2.14	2.48	0.32	0.474
SOD/(U/mL)	23.47	23.81	1.09	0.824
GSH-PX/(U/mL)	334	347	16	0.566
MDA/(nmol/mL)	2.60	2.65	0.15	0.856

**Table 4 biology-12-00441-t004:** Effect of dietary supplementation with stimbiotics on plasma immunoglobulin of weaned piglets.

Items	Treatment		SEM	*p*-Value
	CT	VP		
Day 14				
IgA, mg/mL	2.43	2.97	0.17	0.041
IgG, mg/mL	32.47	35.88	1.02	0.033
Day 42				
IgA, mg/mL	2.32	2.83	0.15	0.037
IgG, mg/mL	38.71	43.76	1.58	0.041

**Table 5 biology-12-00441-t005:** Effect of dietary supplementation with stimbiotics on fecal microbiota α diversity of weaned piglets.

Items	Treatment		SEM	*p*-Value
	CT	VP		
Ace index	497	486	21	0.958
Chao index	497	486	20	0.958
Sobs index	497	485	20	0.958
Shannon index	3.64	4.03	0.17	0.958
Simpson index	0.10	0.06	0.02	1.000

## Data Availability

The datasets generated and/or analysed during the current study are available (NCBI SRA data: PRJNA938621).

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
