# Peer review of "Stimbiotics Supplementation Promotes Growth Performance by Improving Plasma Immunoglobulin and IGF-1 Levels and Regulating Gut Microbiota Composition in Weaned Piglets"

_biology, 2023, doi:10.3390/biology12030441_

Round 1

Reviewer 1 Report

In this study, the authors observed that stimbiotics altered the levels of IGF-1 and the immunity in the plasma, as well as regulated the microbial community in the feces, thus, promoted the growth performance of weaned piglets. This study was well conducted and written, which could provide an interesting option to pig production. However, there are few issues need to be addressed.

1.      The authors only analyzed the IGF-1 levels, thus it is not suitable to describe as “hormone level”.

2.      The abbreviation of “VP group” should be introduced when it first appears in the text, the same for others abbreviations.

3.      Since stimbiotics were a blend of g β-1,4-endo-xylanase and xylo-oligosaccharides, I think the authors should mention the benefits of stimbiotics compared to β-1,4-endo-xylanase or xylo-oligosaccharides in the section of Introduction or Discussion.

4.      Only one dose (100 g/t) of STB was fed to the VP group in this study, the authors should provide the reasons why chose this dose.

5.      The sign “*” indicates the degree of significant difference, thus (* P < ?) should be added in Figure 1.

6.      The conclusion seems too short, and the significance of this study should be added.

Author Response

Response to Reviewer 1:

In this study, the authors observed that stimbiotics altered the levels of IGF-1 and the immunity in the plasma, as well as regulated the microbial community in the feces, thus, promoted the growth performance of weaned piglets. This study was well conducted and written, which could provide an interesting option to pig production. However, there are few issues need to be addressed.

Response (R): Thank you very much for your comments. We have modified the manuscript according to all of your comments.

Comment 1. The authors only analyzed the IGF-1 levels, thus it is not suitable to describe as hormone level.

R: Accordingly, we have revised “hormone level” to “IGF-1 levels” in line 3.

Comment 2. The abbreviation of VP group should be introduced when it first appears in the text, the same for others abbreviations.

R: Accordingly, we have given the full name for “VP” as “VistaPros (VP)” in line 25.

Comment 3. Since stimbiotics were a blend of g β-1,4-endo-xylanase and xylo-oligosaccharides, I think the authors should mention the benefits of stimbiotics compared to β-1,4-endo-xylanase or xylo-oligosaccharides in the section of Introduction or Discussion.

R: Accordingly, we have mentioned the role of β-1,4-endo-xylanase and xylo-oligosaccharides in lines 50-55 and the ability of stimbiotics to combine the effects of both in lines 57-60 of our revised manuscript.

Comment 4. Only one dose (100 g/t) of STB was fed to the VP group in this study, the authors should provide the reasons why chose this dose.

R: Thank you very much for the correction. Some relevant studies have been done on stimbiotics in the previous period to screen out this dose.

Comment 5. The sign “*” indicates the degree of significant difference, thus (* P < ?) should be added in Figure 1.

R: Accordingly, we have added "* P < 0.05" to Figure 1.

Comment 6. The conclusion seems too short, and the significance of this study should be added.

R: Accordingly, we have added the conclusion based on your comments as “The growth performance of weaned piglets improved significantly after STB was added to the diet, probably due to improved immunity and IGF-1 levels in piglets.Because STB supplementation provides piglets not only with enzymes that degrade non-starch polysaccharides but also with prebiotics that benefit intestinal health and immunity, which are used by microorganisms in the intestine to exert helpful effects on the piglets. This is significant for piglets to utilize the nutrients in the feed and improve growth performance.” in lines 364-370.

Reviewer 2 Report

The manuscript „Stimbiotics supplementation promotes growth performance by improving plasma immunoglobulin and hormone levels and regulating gut microbiota composition in weaned piglets” describes the effects of a feed additive and growth performance, fecal microbiota and plasma parameters of weaned piglets. In general, this is an interesting topic, however, I think that the authors overinterpreted some of their results and that they have to discuss the potential benefits and the applied measurements more critically. This applies mainly for the impact of the feeding on growth performance (see below regarding FCR), the meaningfulness of plasma IgA levels and fecal microbiota samples/composition. Moreover, the chaotic use of the term NSPase at the beginning raised the question whether the authors had translation problems or a general misunderstanding of their own research topic.

There are some language issues, therefore I would advise proof-reading of the manuscript by a native speaker.

Moreover, the authors need to upload their sequencing data in a data repository.

Specific comments:

Simple Summary

Line 17: …,weaned, replace greatly by highly

Abstract:

Line 22 introduce the abbreviation body weight (BW), introduce the abbreviation CT for the control group

Line 25 introduce the abbreviation VP

Introduction

Line 38: I would use the term “diet”, not “diet form”

Line 39: in animals the mammary gland is called udder and the term “breast milk” is uncommon, I would recommend replacing it by “sow milk” to emphasize the distinction to milk replacer

Line 41: if the diet would be indigestible, it would not be fed – replace that by “of lower digestibility”

The first paragraph may lead the reader to the conclusion that piglets should simply not be weaned as the feed they are offered then is negatively affected them, which is not true and no alternative, please rephrase this.

Line 43 and following: the authors need to distinguish between non-starch polysaccharides and non-starch polysaccharide degrading enzymes. In the cited references (e.g. reference 12) and in common language the enzymes are referred to by adding -ase(s), so the compounds would be called NSP (non-starch polysaccharides) and the enzymes NSPases. Although the authors are free to define their own abbreviations, later (Line 60) the authors also use the term NSP, and they also refer to xylanase as NSPase, so I believe they mixed up the terms leading to confusion for the reader. The authors should control and correct the terms in the whole manuscript.

Line 47: what is meant by “digestive tract surimi”?

Line 47: “alter” not “alters”

Line 48: “reduce”

Line 49: “NSP are” (see above) …”NSP components”

Line 51: “It has been reported”

Line 53: Make clear that the degradation is achieved through enzymes e.g. xylanase. These enzymes may also be present in some bacterial groups present in the gut microbiota. Please include this.

Line 60: remove “more”

Line 65: NSP

Line 67-71: why did you assume a link to immunity and antioxidant capacity? Please explain

Methods:

Line 75: remove “of”

How many litters were used for achieving the number of 80 piglets? To minimize effects of genetic and sow factors it would be ideal to spread littermates equally into both groups, did you take these factors into account?

Line 77/78: please introduce the abbrevations for the groups

Line 100: when the samples were collected by rectal massage, the authors collected individual samples and pooled them afterwards. Is that correct?

Line: 104/105: was the feed consumption only calculated on those 3 days? The data would be reliable if the feed consumption was evaluated each day and then averaged for the periods.

Line 108: “were” instead of “are”, “was” instead of “is”

Line 116-126: Please improve the language of this paragraph.

Line 151: “pured”

Line 153: nowadays it is more common to use a higher threshold (99%) to build OTUs or to use amplicon sequence variants. Why did you not use a more exact method as your sequencing product is already quite long?

Please give inter- and intra-assay coefficients of variation for ELISA and other photometric measurements.

Results

Table 3: As already mentioned in the methods section I need clarification if the feed consumption was really only analyzed for 1 day per period and the used to calculate the ADFI- then these results are not really reliable

Line 195: IGF-1 is not a plasma immunoglobulin, it is a hormone which is part of the somatotropin axis, the heading needs to be changed.

Table 4: Correct the caption of the table

Table 5: Correct the caption of the table

Line 229-232: Did you assess the differences statistically, then please state the method and the p-values. It is not clear whether these differences are really present.

Line 245: did you correct the p-values for multiple comparisons and by which method? In the text the authors stated that they used the Kruskal wallis rank sum test, however, the figures are labelled as Wilcoxon rank-sum test, so it is not clear which statistical method was used.

Discussion:

Line 269: The authors state that the STB treatment improved the growth performance of the piglets, this is based on their higher feed intake (please consider what it mentioned above regarding this measurement) and the higher weight gain, however the feed conversion rate it statistically not different and numerically lower in the VP group. However, the farmer has to take feed cost into account compared to feed benefits – so why should he pay for an expensive supplement when the FCR which tells him how much money he has to spend on feed per kg of weight gain is not improved? The authors have to discuss all of the parameters here.

Line 270: Why did the authors not also include measuring parameters of fibre fermentation, e.g. by analysis fibre fraction of the feces.

Line 288: IgA is a mucosal immunoglobulin, its plasma concentration is not correlated with its local concentration in the gut or other mucosal surfaces and therefore this parameter is useless as an indicator of (gut) immune function. Moreover, the gut, especially the ileum has its own associated lymphatic tissue (the so called Peyers plates) which is not taken into account here. Therefore, the link of immunity, diarrhea and SCFA here is ambitious and vague. The authors should rewrite this section of the discussion, also because they are repeatingly refering to the beneficial effects of elevated SCFA levels without measuring SCFA levels in their experiments.

Line 316: The statistical differences found in the microbial analysis were mainly attributed to groups with a low relative abundance (Fig. 4B), moreover, the fecal microbial community can differ considerably from the community in the large intestines and even within the gut, the microbial community differs throughout the different small and large intestines. Feces samples are often used due to the easier accessibility, however, the authors have to be caution which conclusion they draw, this should be discussed here.

Conclusion:

The authors should modify their conclusion statement based on the comments above.

Data Availability:

Sequencing data needs to be uploaded on an open access data repository (e.g. NCBI SRA) and the accession number has to be added here.

Reference 12: It is not apparent that this reference is a book chapter, please correct the citation.

Author Response

Response to Reviewer 2:

The manuscript “Stimbiotics supplementation promotes growth performance by improving plasma immunoglobulin and hormone levels and regulating gut microbiota composition in weaned piglets describes the effects of a feed additive and growth performance, fecal microbiota and plasma parameters of weaned piglets. In general, this is an interesting topic, however, I think that the authors overinterpreted some of their results and that they have to discuss the potential benefits and the applied measurements more critically. This applies mainly for the impact of the feeding on growth performance (see below regarding FCR), the meaningfulness of plasma IgA levels and fecal microbiota samples/composition. Moreover, the chaotic use of the term NSPase at the beginning raised the question whether the authors had translation problems or a general misunderstanding of their own research topic. There are some language issues, therefore I would advise proof-reading of the manuscript by a native speaker. Moreover, the authors need to upload their sequencing data in a data repository.

Response (R): Thank you very much for the careful evaluation and thoughtful comments to improve the quality of our manuscript, and we revised the manuscript according to the comments in the whole text.

Specific comments:

Simple Summary

Comment 1. Line 17: , weaned, replace greatly by highly

R: Accordingly, we have revised “greatly” to “highly” in line 17.

Abstract:

Comment 2. Line 22 introduce the abbreviation body weight (BW), introduce the abbreviation CT for the control group

R: Accordingly, we have given the full name for “BW” and “CT group” as “body weight (BW)” and control (CT) group in line 25, respectively.

Comment 3. Line 25 introduce the abbreviation VP

R: We have given the full name for “VP” as “VistaPros (VP)” in line 25.

Introduction

Comment 4. Line 38: I would use the term “diet”, not “diet form”

R: We have revised “diet form” to “diet” in line 38.

Comment 5. Line 39: in animals the mammary gland is called udder and the term “breast milk” is uncommon, I would recommend replacing it by “sow milk” to emphasize the distinction to milk replacer

R: We have revised “breast milk” to “sow milk” in line 39.

Comment 6. Line 41: if the diet would be indigestible, it would not be fed replace that by “of lower digestibility”

R: This was removed due to a revision of the content.

Comment 7. The first paragraph may lead the reader to the conclusion that piglets should simply not be weaned as the feed they are offered then is negatively affected them, which is not true and no alternative, please rephrase this.

R: We modified the manuscript according to your corrections and comments “During the early weaning period, the immaturity of piglets’ organs, coupled with the change in diet, nutritional sources, environment, and psychology, can cause strong stress reactions in piglets. Among them, piglets’ dietary habits transfer from sow milk to corn-soybean meal-based diets. The presence of some anti-nutritional factors in the diet can have an impact on the growth performance of weaned piglets.” in lines 37-41 of our revised manuscript.

Comment 8. Line 43. and following: the authors need to distinguish between non-starch polysaccharides and non-starch polysaccharide degrading enzymes. In the cited references (e.g. reference 12) and in common language the enzymes are referred to by adding -ase(s), so the compounds would be called NSP (non-starch polysaccharides) and the enzymes NSPases. Although the authors are free to define their own abbreviations, later (Line 60) the authors also use the term NSP, and they also refer to xylanase as NSPase, so I believe they mixed up the terms leading to confusion for the reader. The authors should control and correct the terms in the whole manuscript.

R: Accordingly, the abbreviations for non-starch polysaccharides have been standardized to NSP throughout this paper.

Comment 9. Line 47: what is meant by “digestive tract surimi”?

R: The word "digestive tract surimi" has been corrected to "digestive diet" in line 46.

Comment 10. Line 47: “alter” not “alters”

R: We have revised “alters” to “alter” in line 46.

Comment 11. Line 48: “reduce”

R: We have revised “reduces” to “reduce” in line 46.

Comment 12. Line 49: “NSP are” (see above) “NSP components”

R: The abbreviations for non-starch polysaccharides have been standardized to NSP throughout this paper in lines 48-49.

Comment 13. Line 51: “It has been reported”

R: We have revised “It has reported that” to “It has been reported” in line 50.

Comment 14. Line 53: Make clear that the degradation is achieved through enzymes e.g. xylanase. These enzymes may also be present in some bacterial groups present in the gut microbiota. Please include this.

R: The description in the article is not accurate. Since non-ruminants do not produce endogenous enzymes capable of digesting NSP, most of the current research on pigs has been done by adding exogenous feed enzymes to degrade non-starch polysaccharides in feed, and the one that works better is xylanase. The text has been modified as “It has been reported that exogenous addition of non-starch polysaccharide degrading enzymes (especially xylanase, XYL) can degrade insoluble and large soluble polysaccharides in the diet to produce oligosaccharides, oligosaccharides can be well fermented by intestinal bacteria” in lines 50-53 of our revised manuscript.

Comment 15. Line 60 remove “more”

R: It has been removed.

Comment 16. Line 65: NSP

R: We have revised “NSPase” to “NSP” in line 65.

Comment 17. Line 67-71: why did you assume a link to immunity and antioxidant capacity? Please explain

R: Previous studies have demonstrated that xylanase and oligosaccharides can improve intestinal health and reduce oxidative stress [1], and the literature also suggests that oligosaccharides have immunomodulatory effects [2], so in the present study we verify whether STB has similar effects.

[1] Petry, A.L.; Huntley, N.F.; Bedford, M.R.; Patience, J.F. Xylanase increased the energetic contribution of fiber and improved the oxidative status, gut barrier integrity, and growth performance of growing pigs fed insoluble corn-based fiber. J Anim Sci 2020, 98.

[2] De Maesschalck, C.; Eeckhaut, V.; Maertens, L.; De Lange, L.; Marchal, L.; Nezer, C.; De Baere, S.; Croubels, S.; Daube, G.; Dewulf, J.; et al. Effects of Xylo-Oligosaccharides on Broiler Chicken Performance and Microbiota. Appl Environ Microbiol 2015, 81, 5880-5888.

Methods:

Comment 18. Line 75: remove “of”

R: It has been removed.

Comment 19. How many litters were used for achieving the number of 80 piglets? To minimize effects of genetic and sow factors it would be ideal to spread littermates equally into both groups, did you take these factors into account?

R: We selected about 80 piglets completely random from 12 litters, all sows were Landrace-Yorkshire binary sows, but we did not take into account the effects of genetic and sow factors at that time, which we will consider in our future study.

Comment 20. Line 77/78: please introduce the abbrevations for the groups

R: We have given the full name for “CT group” and “VP group” as “control (CT) group” and “VistaPros (VP) group” in lines 77/80, respectively.

Comment 21. Line 100: when the samples were collected by rectal massage, the authors collected individual samples and pooled them afterwards. Is that correct?

R: The description in the article is not accurate. The text has been modified as “On day 42, faecal samples (approximately 20 g) were collected from one randomly selected pig from each pen by rectal massage.” in lines 101-102.

Comment 22. Line: 104/105: was the feed consumption only calculated on those 3 days? The data would be reliable if the feed consumption was evaluated each day and then averaged for the periods.

R: We recorded dietary consumption for each period from days 0 to days 14, 28, and 42, taking the average feed consumption for each period rather than for days 14, 28, and 42. We have described in the Materials and Methods in lines 107-109.

Comment 23. Line 108: “were” instead of “are”, “was” instead of “is”

R: Accordingly, we have revised “are” and “is” to “were” and “was” in lines 111-112

Comment 24. Line 116-126: Please improve the language of this paragraph.

R: The text has been modified as “The plasma antioxidant indices include glutathione peroxidase (GSH-Px), superoxide dismutase (SOD), catalase (CAT), and malondialdehyde (MDA). Using 5,50-dithiobis-p-nitrobenzoic acid to analyze GSH-PX with an absorption peak at 412 nm. SOD activity was measured by the WST-1 method, which uses a water-soluble thiazole salt (WST-1) to react with a superoxide anion to produce a water-soluble dye, thereby measuring SOD activity with an absorption peak at 450 nm. We measured CAT activity with ammonium molybdate and its absorption peak was then 405 nm. Measurement of MDA activity by the TBA method, which condenses MDA with 2-thiobarbituric acid to form a red product with an absorption peak at 532 nm. The above kits were purchased from Nanjing Jiancheng Bioengineering Institute (Nanjing, China), and the determination was carried out according to the kit instructions. ” in lines 120-130.

Comment 25. Line 151: “pured”

R: We have revised “pureed” to “pure” in line 155.

Comment 26. Line 153: nowadays it is more common to use a higher threshold (99%) to build OTUs or to use amplicon sequence variants. Why did you not use a more exact method as your sequencing product is already quite long?

R: The description in the text was incorrect. We are analyzing microbial data based on 99% of ASV levels. We have corrected it as “Categorize unique sequences into identical amplicon sequence variants (ASVs) by a 99% similarity threshold. Statistical analysis was performed using the ASV agglomerative results. ” in line 157.

Comment 27. Please give inter- and intra-assay coefficients of variation for ELISA and other photometric measurements.

R: We have given inter- and intra-assay coefficients of variation for ELISA as The intra-assay and the inter-assay coefficients of variation for ELISA in the samples were less than 10% and 15%, respectively.” in lines 142-144.

Results

Comment 28. Table 3: As already mentioned in the methods section I need clarification if the feed consumption was really only analyzed for 1 day per period and the used to calculate the ADFI- then these results are not really reliable.

R: Thank you very much for your suggestion. These results are reliable. Because we did not analyze only 1 day of feed consumption, we recorded ration consumption from days 0 to 14, 28, and 42 days for the calculation of ADFI.

Comment 29. Line 195: IGF-1 is not a plasma immunoglobulin, it is a hormone which is part of the somatotropin axis, the heading needs to be changed.

R: The heading has been revised “Plasma immunoglobulin levels” to “Plasma immunoglobulin and IGF-1 levels” in line 198.

Comment 30. Table 4: Correct the caption of the table

R: The caption of the table has been revised to “Table 4. Effect of dietary supplementation with stimbiotics on plasma immunoglobulin of weaned piglets” in line 205.

Comment 31. Table 5: Correct the caption of the table

R: The caption of the table has been revised to “ Table 5. Effect of dietary supplementation with stimbiotics on fecal microbiota α diversity of weaned piglets ” in line 229.

Comment 32. Line 229-232: Did you assess the differences statistically, then please state the method and the p-values. It is not clear whether these differences are really present.

R: The description in the text was incorrect, we have removed the relevant content.

Comment 33. Line 245: did you correct the p-values for multiple comparisons and by which method? In the text the authors stated that they used the Kruskal wallis rank sum test, however, the figures are labelled as Wilcoxon rank-sum test, so it is not clear which statistical method was used.

R: The description in the text was incorrect and it has been revised to the Wilcoxon rank sum test in line 250.

Discussion:

Comment 34. Line 269: The authors state that the STB treatment improved the growth performance of the piglets, this is based on their higher feed intake (please consider what it mentioned above regarding this measurement) and the higher weight gain, however the feed conversion rate it statistically not different and numerically lower in the VP group. However, the farmer has to take feed cost into account compared to feed benefits so why should he pay for an expensive supplement when the FCR which tells him how much money he has to spend on feed per kg of weight gain is not improved? The authors have to discuss all of the parameters here.

R: Accordingly, we have added to the content “It is noteworthy that there was no significant difference in FCR, which may be due to the short period of our experiment and the change in FCR was not reflected. But the IGF-1 levels in piglets were improved, which was consistent with the change in body weight, daily weight gain and daily feed intake of weaned piglets.” in lines 276-280.

Comment 35. Line 270: Why did the authors not also include measuring parameters of fibre fermentation, e.g. by analysis fibre fraction of the feces.

R: We did not add the relevant content when designing the experimental protocol because previous studies have demonstrated that STB can promote fiber fermentation, and we wanted to study the effect from another perspective, so we did not measure the parameters related to fiber fermentation.

Comment 36. Line 288: IgA is a mucosal immunoglobulin, its plasma concentration is not correlated with its local concentration in the gut or other mucosal surfaces and therefore this parameter is useless as an indicator of (gut) immune function. Moreover, the gut, especially the ileum has its own associated lymphatic tissue (the so called Peyers plates) which is not taken into account here. Therefore, the link of immunity, diarrhea and SCFA here is ambitious and vague. The authors should rewrite this section of the discussion, also because they are repeatingly refering to the beneficial effects of elevated SCFA levels without measuring SCFA levels in their experiments.

R: Thank you for your comment. We agree with your opinion. Accordingly, we have made some changes to the content “The mucosal immune system is not fully mature when piglets are weaned, are also encounter psychosocial, physical and nutritional stressors that can have a negatively cumulative effect on the immune response, resulting in lagging growth after weaning. It has been demonstrated that the plasma IgA and IgG concentrations will decrease directly after weaning [25,26], causing severe weaning stress [27]. The main functions of IgA and IgG are respectively to protect mucosal surfaces from infectious microorganisms [28] and recognize antigens on the surface of invading viruses and bacteria and recruit effector molecules [29], playing a key role as antibodies against infection in the immune system [30-32]. Our study demonstrated that weaned piglets supplemented with STB had substantially higher levels of immunoglobulins (IgA and IgG) than the CT group, suggesting that dietary STB can improve the immune system of piglets. However, STB had no obvious effect on reducing the rate of diarrhoea in weaned piglets, and the main reason might be related to the addition of ZnO to the pre-stater diet. It is widely known that feeding ZnO for 14 consecutive days after weaning is effective in preventing and reducing the incidence of diarrhoea in weaned piglets [33]. Therefore, low diarrhoea incidence was obtained in the two groups during the early nursery period. ” in lines 292-308.

Comment 37. Line 316: The statistical differences found in the microbial analysis were mainly attributed to groups with a low relative abundance (Fig. 4B), moreover, the fecal microbial community can differ considerably from the community in the large intestines and even within the gut, the microbial community differs throughout the different small and large intestines. Feces samples are often used due to the easier accessibility, however, the authors have to be caution which conclusion they draw, this should be discussed here.

R: Accordingly, we have made a change to the content “Unfortunately, since fecal samples are more readily available, we only collected feces for microbial analysis, and the fecal microbial community may be very different from that in the colon or even the gut, also the relative abundance of these differential microorganisms is low, so it cannot be verified that all of these microorganisms are at play, so more specific studies will be conducted later.” in lines 335-341.

Conclusion:

Comment 38. The authors should modify their conclusion statement based on the comments above.

R: Accordingly, we have made a change to the conclusion based on your comments “The growth performance of weaned piglets improved significantly after STB was added to the diet, probably due to improved immunity and IGF-1 levels in piglets.Because STB supplementation provides piglets not only with enzymes that degrade non-starch polysaccharides but also with prebiotics that benefit intestinal health and immunity, which are used by microorganisms in the intestine to exert helpful effects on the piglets. This is significant for piglets to utilize the nutrients in the feed and improve growth performance.” in lines 364-370.

Data Availability:

Comment 39. Sequencing data needs to be uploaded on an open access data repository (e.g. NCBI SRA) and the accession number has to be added here.

R: The sequencing data has been uploaded to NCBI and added the accession number PRJNA938621 in line 384.

Comment 40. Reference 12: It is not apparent that this reference is a book chapter, please correct the citation.

R: Accordingly, we have modified the reference 12 as “Gonzalez-Ortiz, G.; Gomes, G.A.; dos Santos, T.T.; Bedford, M.R. In The Value of Fibre-Engaging the Second Brain for Animal Nutrition.; 2019; pp. 233-254.” in lines 420-421.